# Primer extension coupled with fragment analysis for rapid and quantitative evaluation of 5.8S rRNA isoforms

**Giulia Venturi[1,2], Federico Zacchini[1,2], Cinzia Lucia Vaccari[1,2], Davide Trerè[1,3], Lorenzo Montanaro[1,2,3]** *

**1** Dipartimento di Medicina Specialistica, Diagnostica e Sperimentale (DIMES), Alma Mater Studiorum—Università di Bologna, Bologna, Italia, **2** Centro di Ricerca Biomedica Applicata–CRBA, Università di Bologna, Policlinico di Sant'Orsola, Bologna, Italia, **3** Programma Dipartimentale di Medicina di Laboratorio, IRCCS Azienda Ospedaliero-Universitaria di Bologna, Bologna, Italia

* lorenzo.montanaro@unibo.it

**Data Availability Statement:** All relevant data are within the paper.

## Abstract

The ribosomal RNA 5.8S is one of the four rRNAs that constitute ribosomes. In human cells, like in all eukaryotes, it derives from the extensive processing of a long precursor containing the sequence of 18S, 5.8S and 28S rRNAs. It has been confirmed also in human cells the presence of three isoforms of 5.8S rRNA: one more abundant called 5.8S short, one called 5.8S long bearing 5 extra-nucleotides at its 5' end and one 10 nucleotide shorter called 5.8S cropped. So far, little is known about 5.8S long specific role in cell biology and its function in human pathology. The lack of studies on the three 5.8S isoforms could be due to the techniques usually applied to study ribosome biogenesis, such as Northern blot with radioactively labelled probes, that require strict protective measures, and abundant and high-quality samples. To overcome this issue, we optimized a method that combines primer extension with a fluorescently labeled reverse primer designed on the 3' of 5.8S rRNA sequence and fragment analysis. The resulting electropherogram shows the peaks corresponding to the three isoforms of 5.8S rRNA. The estimation of the area underneath the peaks allows to directly quantify the isoforms and to express their relative abundance. The relative abundance of 5.8S long and 5.8S short remains constant using scalar dilution of RNA and in samples subjected to partial degradation. 5.8S cropped abundance varies significantly in lower concentrate RNA samples. This method allows to analyze rapidly and safely the abundance of 5.8S rRNA isoforms in samples that have been so far considered not suitable such as poorly concentrated samples, RNA derived from frozen tissue or unique samples.

## Introduction

5.8S ribosomal RNA (rRNA) is one of the four RNA species that compose eukaryotic ribosomes. It derives from a highly coordinated biogenesis process that starts in the nucleolus with the transcription of a long precursor followed by extensive editing and a number of sequential cleavages (see Box 1 for a schematic representation of the key steps of rRNA processing) [1].

**Funding:** This work was funded by Fondazione AIRC to LM (grant number IG 21562 - www.airc.it) and by Bologna University funds (www.unibo.it) from the Pallotti Legacy for Cancer Research to LM (no grant number available). The funders had no role in study design, data collection and analysis, decision to publish, or preparation of the manuscript.

**Competing interests:** The authors have declared that no competing interests exist.

## Box 1. Schematic representation of ribosomal RNA processing

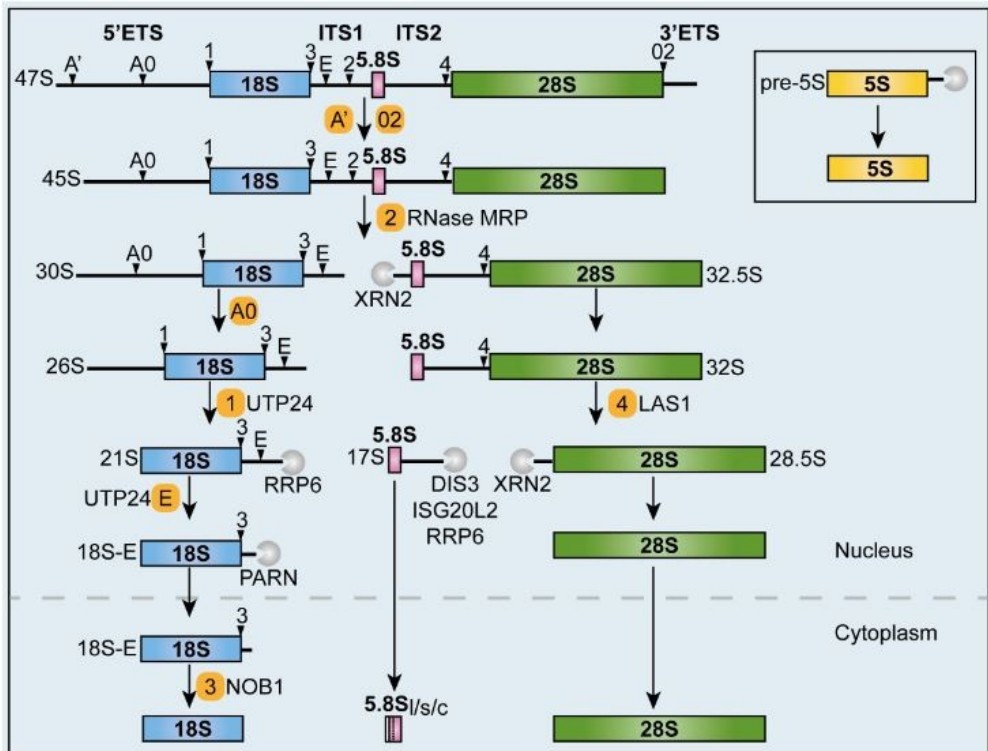

Pre-ribosomal RNA processing in mammalian cells starts in the nucleolus with the transcription, operated by RNA polymerase I, of a long precursor (47S) from the numerous repeats of rDNA genes. The 47S pre-rRNA is then extensively cleaved and modified to originate the mature 18S, 28S and 5.8S rRNA (widely reviewed in [11]) while 5S rRNA follows a different pathway. In the 47S precursor 18S, 5.8S and 28S rRNA are flanked by two external transcribed spacer (5' and 3' ETS) and separated by two internal transcribed spacers (ITS1 and ITS2). An initial cleavage at site A' in the 5'ETS and at site 2 in the 3'ETS originates 45S pre-rRNA that is processed following alternative pathways either starting in the 5'ETS or in the ITS1 [12]. The major pathway described in HeLa cells starts with the endonucleolytic cleavage at site 2 of ITS1 [13] and leads to the formation of 30S rRNA and 32.5S rRNA. From this point the two precursors will follow separated pathways. In fact, 30S rRNA is cleaved at site A0 and 1 originating 21S that is processed by 3'-5' trimming [14]. The product then undergoes cleavage at site E [15] and 3'-5' trimming by PARN before cytoplasmic export [16]. In the cytoplasm 18S rRNA is obtained by cleavage at site 3 by NOB1 [14]. In parallel, 32.5S rRNA is processed through 5'-3' exonucleolytic cleavage at its 5' end by XRN2 originating in this way the two variants of the 5.8S rRNA 5' end [17], followed by LAS1 action on site 4 that separates 5.8S and 28S precursors [18]. Eventually, before cytoplasmic export, 5.8S precursor is progressively trimmed at the 3' end by DIS3, ISG20L2 and RRP6 exonucleases while the 5' end of 28S rRNA is tailored by XRN2 [11, 17].

Ribosome biogenesis has been widely studied in yeast and only recently several studies have been conducted on human cell lines. Three isoforms of 5.8S rRNA have been described: a long form (5.8S long—l -), a shorter one missing 5 nucleotides at the 5' of the sequence (5.8S short—s -) [2, 3] and a 5'-truncated isoform (5.8S cropped—c-) [4]. One of the current models for 5.8S short biogenesis involves the cleavage at the internal transcribed spacer 1 (ITS1) operated by RNAse MRP and the following exonucleolytic 5'-3' trimming [5–7]. However, there are reports challenging this viewpoint, indicating that ITS1 cleavage could be RNase MRP-independent [8]. On the other hand, at least in yeast, the long isoform derives from direct ITS1 cleavage by a still uncharacterized enzyme [9]. A specific role for the 5.8S long isoform has not been characterized. This is probably due to technical difficulties that researchers face when studying ribosomal RNA. In fact, the most used technique to analyze rRNAs and their processing is Northern blot using probes labelled with radioactive tag. This technique is highly useful to study rRNA biogenesis since it allows to determine the size of rRNA intermediates and even small changes derived by their processing. However, Northern blot cannot be applied on a large systematic scale due to the use of hazardous reagents, such as radioactive probes, that require dedicated rooms and strict protective measures. Recently, chemiluminescent probes have been introduced to reduce the problems derived from radioactive reagents, but their use reduces the sensitivity of the method. On the other hand, due to their inherent methodological features, structural approaches also failed to characterize the properties of 5.8S long isoform [10]. To overcome these issues, we propose to exploit primer extension and fragment analysis to study 5.8S rRNA isoforms and their relative representation. In fact, the method we optimized permits a rapid, safe and easy quantification of the three 5.8S isoforms, allowing their analysis in a wide range of possible samples, including difficult and poorly concentrated samples.

## Materials and methods

### Cell culture

HeLa and U2OS cell lines were cultured in Dulbecco's Modified Eagle's Medium (DMEM) with high glucose supplemented with 10% FBS, 2 mM L-Glutamine, 100 U/ml Penicillin and 100 µg/ml Streptomycin (all provided by EuroClone) at 37°C in a humidified incubator with 5% $CO_2$.

### RNA extraction

Total RNA was extracted from cells at 70–80% confluence using Purezol reagent (BioRad) following manufacturer's instruction. RNA concentration was assessed using NanoDrop (Life Technologies) at 260 nm.

### Ribosome isolation

Ribosomes from U2OS cells were isolated essentially as previously described [19]. Briefly, $1x10^6$ cells were plated in five 150 mm dishes and cultured for 72 hours before lysis. Cell lysis was carried out for 10 minutes on ice using lysis buffer containing 10 mM Tris-HCl pH 7.5, 10 mM NaCl, 3 mM MgCl2 and 0.5% (vol/vol) Nonidet P40 followed by centrifugation at 20 000g for 10 minutes at 4°C. Highly purified ribosomes were pelleted by ultracentrifugation of the cell lysate on a discontinuous sucrose gradient at 120 000g at 4°C for 16 hours and quantified measuring their absorbance at 260 nm. Ribosomal RNA was extracted from 8 pmol of ribosomes using Purezol reagent (BioRad).

## Simulation of RNA degradation

Total RNA from HeLa cells (1.5 μg) was incubated at 90°C for 0, 10, 20, 30, 40 and 50 minutes (adapted from [20]). After incubation half of the sample was run on 1% agarose gel with Ethidium Bromide 0.5 μg/ml in TBE buffer to assess RNA degradation.

## Primer extension and fragment analysis

To perform primer extension, from 0.06 to 1 μg of total RNA from cell lines or Human Xpress-Ref Universal Total RNA (Qiagen) complexed with 10 pmol of 5.8S Rev FAM primer and 2 pmol of 5S Rev Hex primer (Integrated DNA Technologies, HPLC purification) in a total volume of 3 μl was denatured at 92°C for 2.5 minutes and incubated at 59°C for 30 minutes for primer annealing followed by 5 minutes on ice. Then the mix containing water, GoScript Reaction Buffer, 5 mM MgCl$_2$, dNTPs (0.5 mM each), 10 u of RNase inhibitor and 160 u of GoScript Reverse Transcriptase (GoScript Reverse Transcriptase kit Promega) was added to the sample to obtain a final volume of 10 μl and incubated at 42°C for 1 hour. This reaction is carried out in 0.2 ml tubes (Multiply-Pro cup PP Sarstedt) in a thermal cycler (T100 Bio-Rad). After primer extension, 1 μl of sample was added to 8.8 μl of Hi-Di Formamide (Applied Biosystems) and 0.2 μl of GeneScan 500 Liz dye size standard (Applied Biosystems). Samples were separated on 36 cm capillary array using POP-7 Polymer on an Abi 3730 Sequencer (Applied Biosystems) using DS-33 Matrix (Dye Set G5) for spectra calibration.

## Evaluation of 5.8S isoforms and statistical analysis

The fragment analysis data file obtained from the sequencer was analyzed using Peak Scanner software (Thermo Scientific Cloud). We evaluated each single isoform as a ratio between the area underneath the corresponding peak and the sum of the total areas of all the three peaks that can be ascribed to 5.8S (5.8Sc + 5.8Ss + 5.8Sl). Data in Fig 4 were analyzed using SPSS software. We performed Student's T test to compare differences between samples.

## 5'RACE (rapid amplification of cDNA end)

5'RACE (5'/3' RACE kit 2nd generation Roche) was performed using total and ribosomal RNA from U2OS cells following manufacturers' instructions with few modifications to the protocol. We added a denaturation step at 90°C for 2 minutes and an annealing step at 59°C for 30 minutes to the suggested first strand cDNA synthesis protocol using the primer 5.8S Rev (Table 1). Reverse transcription (RT) was carried out at 42°C for 1 hour. First strand cDNA was then purified using PCR purification kit (Qiagen). Poly(A) or poly(T) tailing of the first strand cDNA was then performed. Finally, a PCR using AccuPrime, a high fidelity Taq DNA polymerase (Invitrogen), was performed using 5.8S Rev primer (Table 1) and the degenerated oligo dT-anchor Primer of the kit in the case of poly(A) tailing or a degenerated oligo dA-anchor primer (IDT) in the case of poly(T) tailing. The PCR product was then purified using PCR purification kit (Qiagen) and Sanger sequenced.

**Table 1. Name and sequence of the used primers.**

| Primer name | Primer sequence (5'-3') |
| --- | --- |
| 5.8S Rev FAM | /56-FAM/AAGCGACGCTCAGACAGGCGTA |
| 5.8S Rev | AAGCGACGCTCAGACAGGCGTA |
| 5S Rev HEX | /5HEX/CCTACAGCACCCGGTATTCC CA |
| oligo dA-anchor primer | GACCACGCGTATCGATGTCGACAAAAAAAAAAAAAAA/T/C/G |

## Results

### Evaluation of 5.8S cropped, short and long rRNA using primer extension and fragment analysis

The method described to evaluate 5.8S rRNA isoforms consist in a primer extension using two different primers: one reverse primer conjugated with FAM fluorophore designed to fully reverse-transcribe 5.8S rRNA sequence, and a second reverse primer conjugated with HEX fluorophore that mediates the reverse transcription of 5S rRNA used as internal standard (Table 1). The protocol starts with a denaturation step at 92˚C, important to fully unfold rRNA secondary structures, followed by an annealing step at 59˚C before the RT reaction. The samples are mixed with formamide and size standard, then undergo fragment analysis in which the fluorescently labeled DNA fragments are separated according to their size (Fig 1A). The output consists in an electropherogram showing one HEX fluorescent peak at 114 nucleotides corresponding to 5S rRNA, plus one FAM peak at 141 nucleotides, one more abundant FAM peak at 153 nucleotides and one FAM peak at 158 nucleotides that we identified as 5.8S rRNA cropped, short, and long, respectively (Fig 1B). The analysis of the electropherogram using the software Peak Scanner available at Thermo Fisher Cloud allows to measure the area underneath the peaks used to calculate the ratio between each peak of interest (cropped, short or long) and the sum of the three peaks. Considering how we designed the primers, we noticed

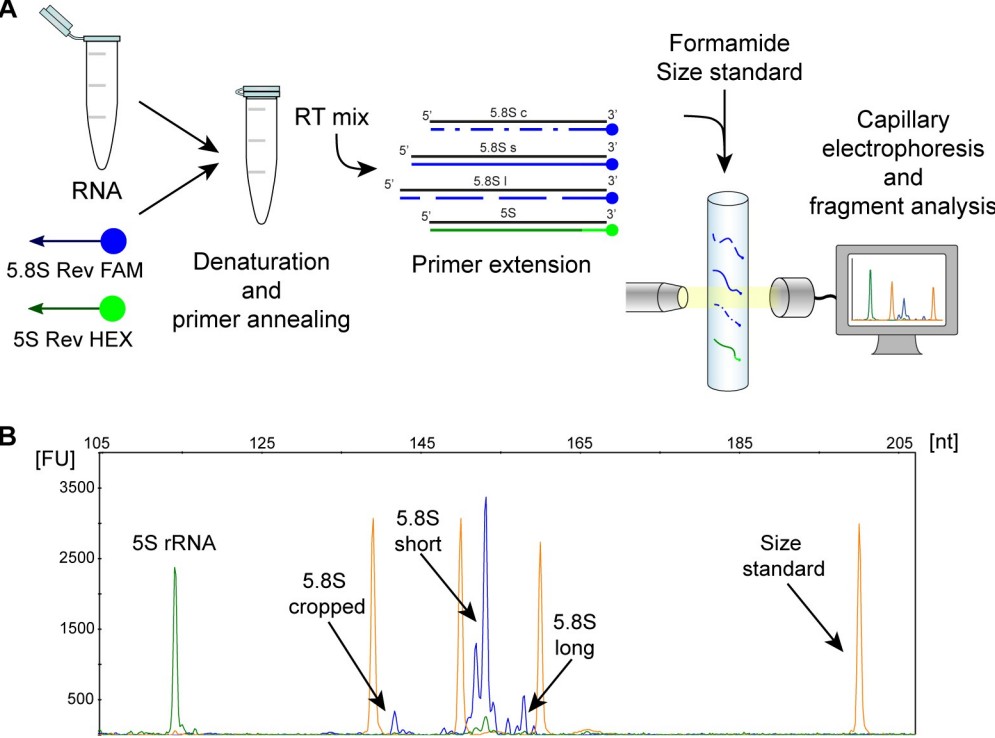

**Fig 1. Schematic representation of the method. A** sample containing RNA, 5.8S Rev FAM and 5S Rev HEX primers is denatured then incubated for annealing. Then, the reverse transcription reaction is performed. An aliquot of the reaction product is complexed with formamide and size standard and run through capillary electrophoresis. The three isoforms are indicated as follows: c stands for 5.8S cropped, s for 5.8S short and l for 5.8S long. **B** The fragment analysis file generated by the sequencer can be analyzed using Peak Scanner Software. Yellow peaks are the size standard, the green peak is the internal reference 5S rRNA, whereas blue peaks are the products of the primer extension reaction. Upper X axis represents the base length of each fragment; Y axis represents the relative fluorescence of each detected fragment.

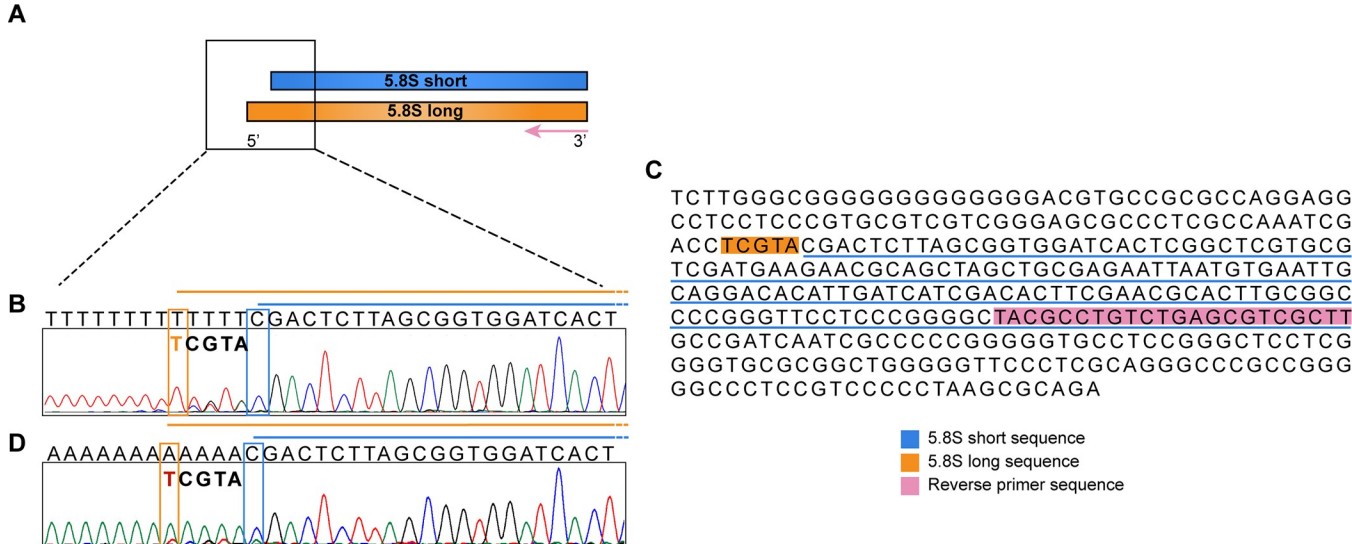

**Fig 2. 5'RACE and sequence of 5.8S short and long isoforms. A** Schematic representation of 5.8S rRNA short and long and 5.8S Rev primer used for 5'RACE. **B** Sequencing of the product obtained from 5'RACE and poly(A) tailing using total RNA. 5.8S short sequence is reported in black on top of the chromatogram while the five extra nucleotide of 5.8S long are in bold. **C** 45S precursor rRNA sequence in which is possible to see in pink the 5.8S Rev primer sequence, in blue the 5.8S short published sequence and in orange the longer sequence of 5.8S long. **D** Sequencing of the product obtained from 5'RACE and poly(T) tailing using ribosomal RNA. We confirmed that the first T (in red) is part of the 5.8S long sequence.

a difference between the expected 5S and 5.8S size, of 117 and 157 nucleotides respectively, and what observed. Since this difference involves both the transcripts analyzed it can be due to the effect of the fluorescent dye on the fragment mobility or to an imperfect alignment between the size standard and the sample. These limited differences in size determination are also expected according to technical approach employed and are not surprising. In addition, the peaks that we identify in the electropherogram appear in the same position in all performed experiments. Thus, although the sizing of the peaks may not be completely accurate, the results can still be considered precise and reproducible if the samples are run in the same conditions.

To confirm that this assay was able to reverse transcribe specifically 5.8S rRNA, we performed a 5' rapid amplification of cDNA ends (RACE) reaction on total and ribosomal RNA using a non-fluorescent 5.8S Rev primer (Fig 2A). The product of the reaction performed on total RNA from U2OS cells underwent Sanger sequencing (Fig 2B). It resulted in a predominant sequence corresponding to the registered 5.8S (NR_003285.3) and in a less abundant sequence four or five nucleotides longer at its 5'. This sequence corresponds to the sequence at the 5' end of 5.8S published sequence in the 45S precursor (Fig 2C) (NR_046235.3) but, due to the used method, the first T was uncertain. Therefore, we performed a second 5' RACE reaction in which we carried out a poly(T) tailing instead of poly(A) tailing. In order to avoid the confounding effect of pre-rRNA present in total RNA, we performed this evaluation solely on mature rRNA, derived from purified ribosomes. The result confirmed that the predominant sequence corresponded to the deposited 5.8S sequence (NR_003285.3) and that a five nucleotides longer sequence (terminating with a T at the 5' end) was also present (Fig 2D). A less represented T peak was also identified in correspondence of the first C at the 5' of 5.8S short sequence in line with the known presence of one nucleotide heterogeneity at the 5' end [4]. In this assay it was not possible to selectively identify the cropped isoform due to its limited quantitative representation with respect to the 5.8S short isoform.

Taken together these results demonstrate that our 5.8S Rev primer is suitable for the reverse transcription of the 5.8S rRNA sequence. In addition, we could also confirm that the longer

isoform contains a sequence of five extra nucleotides (TCGTA) at its 5' end in line with what has been previously reported by Heindl *et al* [2].

## Optimization of RNA and primer amount

To determine the quantity of 5.8S rRNA reverse primer suitable for our assay we performed primer extension with two amounts of primer (1 pmol and 10 pmol) combined with several quantities of total RNA (0.06 to 1 μg) from U2OS cell line (Fig 3A) and we evaluated the area of the 5.8S short peak. Based on the results, we selected 10 pmol primer amount, which ensured a linear correlation between RNA input and 5.8S detection (R square value of the calculated regression line: 0.97 for 10 pmol of primer versus 0.85 for 1 pmol of primer). To optimize 5S reverse primer amount we evaluated 5S peak area corresponding to 1, 2, 5 and 7 pmol of primer in reactions containing 10 pmol of 5.8S primer and total RNA from U2OS cells (Fig 2B). We selected the amount of 5S primer (2 pmol) that allowed to obtain similar area values for 5S peak and 5.8S short peak, while not altering 5.8S reverse transcription reaction (Fig 3C).

## Robustness of the method

To determine the robustness of the method we evaluated the amount of 5.8S isoforms using scalar dilutions of a standard RNA derived from different human tissues (Fig 4A). We

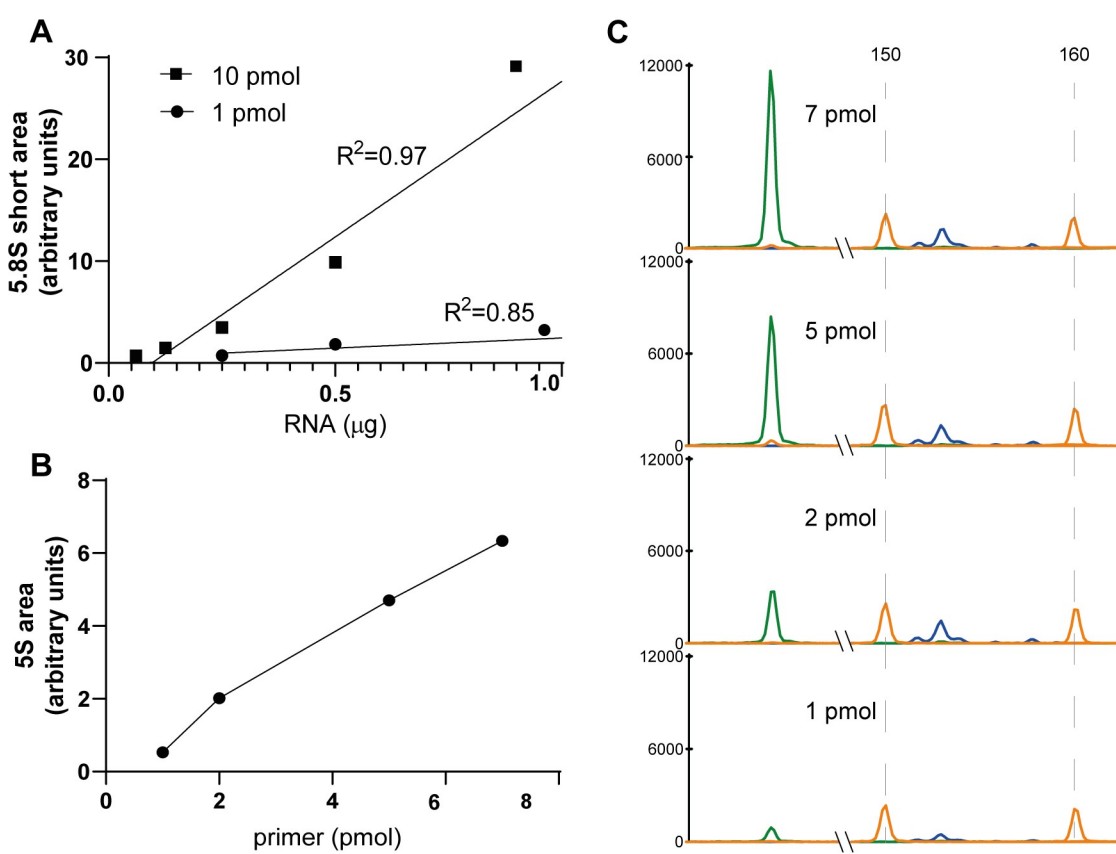

**Fig 3. 5.8S Rev FAM and 5S Rev HEX primers optimization. A** Evaluation of two 5.8S Rev FAM primer quantities, 1 pmol and 10 pmol, using 0.06, 0.125, 0.25, 0.5 and 1 μg of RNA from U2OS cells. The plot reports 5.8S short peak area and R2 values were obtained by linear regression. **B and C** Optimization of 5S Rev HEX primer using 1, 2, 5 and 7 pmol of primer in a reaction with 10 pmol of 5.8S Rev FAM primer and 0.125 μg of total RNA from U2OS cells. The introduction of the second primer did not affect the detection of 5.8S rRNA.

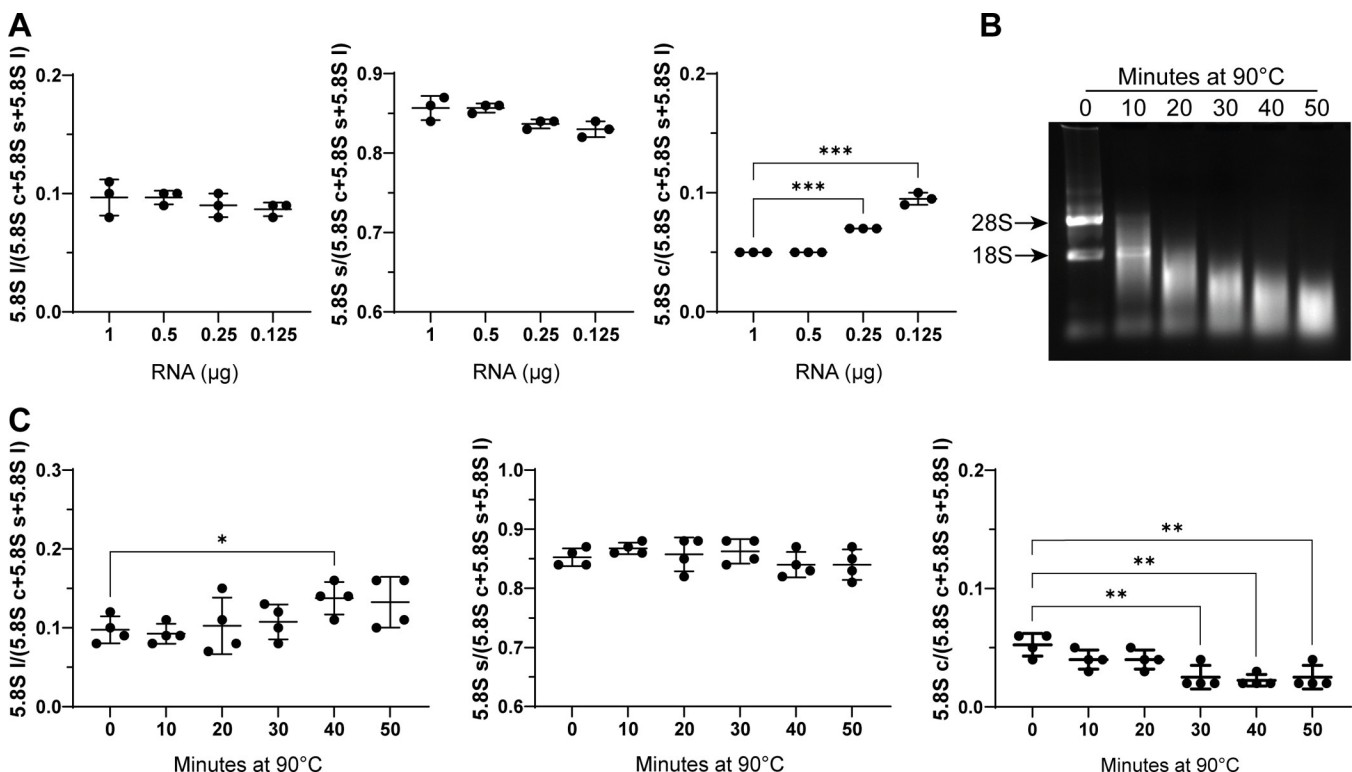

**Fig 4. Robustness of the method. A** Evaluation of 5.8S isoforms using scalar dilutions of standard RNA in triplicate. The average values are consistent in the range of the quantities used except for 5.8Sc whose level remained stable only when using at least 0.5 µg of RNA. Comparison between the higher concentrated samples and each other concentration were performed using Student's T test. **B** Representative electrophoresis image showing RNA degradation in samples incubated from 0 to 50 minutes at 90˚C. **C** Evaluation of 5.8S isoforms using RNA degraded for 0, 10, 20, 30, 40, and 50 minutes at 90˚C. The average values obtained from four replicates is similar from 0 to 20 minutes at 90˚C. After 20 minutes the values of 5.8Sl and 5.8Sc resulted significantly different from the non-degraded samples. Comparison between the undegraded samples and each other condition were performed using Student's T test. *<0.05, **<0.01, ***<0.001.

observed that the average ratio values of 5.8Sl and 5.8Ss remained stable when RNA amount decreased (Fig 4A). However, when we evaluated 5.8Sc, the average ratio value was significantly increased when RNA amount was reduced, indicating that the evaluation of 5.8Sc with this method is more efficient in more concentrated samples. In addition, to assess the state of 5.8S isoforms over sample degradation, we performed primer extension and fragment analysis on total RNA from HeLa cells subjected to heat degradation at 90˚C from 0 to 50 minutes (Fig 4B). Results showed that the average value of each isoform was constant up to 20 minutes. When RNA degradation increased, we observed a rise in 5.8Sl values and a decrease of 5.8Sc values, suggesting a different sensitivity to degradation of the different isoforms (Fig 4C). These results indicate that the method described here can be considered reliable in the tested RNA quantity range and that it can detect efficiently all the three isoforms in partially degraded samples whose aspect is similar to the sample incubated at 90˚C for 20 minutes.

## Discussion

From a literature search, it emerged that recent studies on 5.8S rRNA potential role in healthy and pathological tissues are currently lacking, in particular regarding 5.8S long. This could be due to the fact that in most NGS-based gene expression techniques rRNA is usually excluded from analyses. A second reason for the lack of studies on 5.8S rRNA could be ascribed to the

fact that the most common technique used to study rRNA processing is Northern Blot. In fact, this technique is highly efficient for rRNA intermediates size detection but shows limitations in rRNA quantification. Moreover, it requires high quantities of good quality RNA (typically from 1.5 to 2.5 µg). In addition, given the use of radioactively labeled probes, dedicated rooms and strict protective measures to work with are necessary. On the contrary, the method that we propose, based on primer extension with fluorescent reverse primer designed on the 3' of 5.8S rRNA followed by fragment analysis, results more easy and rapid compared to Northern Blot and it shows higher resolution. The determination of the area underneath the peaks using Peak Scanner software enables a direct quantification of the different peaks, identifying the different isoforms. Moreover, we demonstrated that it can be performed with limited amounts of RNA and with partially degraded samples (Fig 4). These features allow the analysis of 5.8S rRNA in so far excluded samples, such as RNA derived from frozen tissue, precious samples, or poorly concentrated samples.

A further aspect worth of consideration is represented by the heterogeneity of peaks detected with the described technique. In fact, comparing 5S and 5.8S short peaks in Figs 1 and 3C, it is possible to see that they present a different shape. All RNA samples analyzed presented the same pattern that consist in a sharp and thin 5S peak (in green) and a larger 5.8S short peak (in blue) presenting two to three humps: a higher one corresponding to 153 nucleotides plus one 1-base shorter and one 1-base longer, not always detected. This difference can be due to the to the different reverse transcription efficiency on 5.8S rRNA, that presents several secondary structures. In addition, the M-MLV reverse transcriptase used here is known to append non-templated nucleotides to the 3' end of RNA/cDNA duplexes, thus generating products with a random extra nucleotide [21–23]. This kind of event should occur on all 5' uncapped RNA templates present in the reaction not affecting the possibility to discriminate the different RNA isoforms. Nonetheless, since 5' portion of 5.8S originate from the exonucleolytic activity of XRN2 and other enzymes, as reported in literature [5–8], this heterogeneous peak could also represent trimming intermediates. Another possibility to consider, is that because of the previously mentioned one nucleotide heterogeneity at 5' end of 5.8S short [4] additional isoforms of 5.8S could be present in cells, thus increasing the complexity of ribosomes. This possibility is in line with what reported by the studies describing 5.8S isoforms for the first time in human cells [3]. These results may suggest that ribosomes could differ from one another due to the presence of different 5.8S rRNA isoforms contributing, together with other elements, to ribosome heterogeneity. Therefore, these isoforms can be identified and further characterized using this method, given its high resolution. In this sense the method here described could be applied to characterize the 5.8S composition of ribosomes from different tissues (including normal and pathological tissues) or with different functional properties.

## Supporting information

**S1 Raw images.**
(PDF)

## Acknowledgments

The Authors would like to thank Dr. Simona Ferrari for technical help and for the access to the sequencing facility and Dr. Marianna Penzo for critically reading the manuscript.

## Author Contributions

**Conceptualization:** Giulia Venturi, Lorenzo Montanaro.

**Formal analysis:** Davide Trerè.

**Funding acquisition:** Lorenzo Montanaro.

**Investigation:** Giulia Venturi, Federico Zacchini, Cinzia Lucia Vaccari.

**Methodology:** Davide Trerè.

**Supervision:** Lorenzo Montanaro.

**Validation:** Giulia Venturi.

**Writing – original draft:** Giulia Venturi.

**Writing – review & editing:** Federico Zacchini, Davide Trerè, Lorenzo Montanaro.

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
