## [Decision Letter · Decision Letter 0]

25 Oct 2021

PONE-D-21-30501Primer extension coupled with fragment analysis for rapid and quantitative evaluation of 5.8S rRNA isoformsPLOS ONE

Dear Dr. Montanaro,

Thank you for submitting your manuscript to PLOS ONE. After careful consideration, we feel that it has merit but does not fully meet PLOS ONE’s publication criteria as it currently stands. Therefore, we invite you to submit a revised version of the manuscript that addresses the points raised during the review process.

In addition to carefully addressing the concerns and comments raised by the reviewers, please take care of the following issues: 1) Line 104, please specify A260 for RNA quantification.2) Please consider adding a diagram in Fig.1 (possibly as Fig.1B and put current Fig.1B as 1C) to illustrate the mechanism of the method, including the labeling of the 5.8S rRNA (three forms) and the 5S rRNA by respective labeled primers in reverse transcription, the separation of the cDNA, the detection and the quantification.3) Please check sentence 241 for readability.4)Fig4B, line 254, please check whether the ratio decreases or rises.

We look forward to receiving your revised manuscript.

Kind regards,

Baisong Lu, Ph.D

Academic Editor

PLOS ONE

Journal Requirements:

[The Authors would like to thank Dr. Simona Ferrari for technical help and for the access to the sequencing facility and Dr. Marianna Penzo for critically reading the manuscript. This work was funded by Fondazione AIRC to LM (grant number IG 21562 - www.airc.it) and by Bologna University funds (www.unibo.it) from the Pallotti Legacy for Cancer Research to LM (no grant number available). The funders had no role in study design, data collection and analysis, decision to publish, or preparation of the manuscript.]

 [This work was funded by Fondazione AIRC to LM (grant number IG 21562 - www.airc.it) and by Bologna University funds (www.unibo.it) from the Pallotti Legacy for Cancer Research to LM (no grant number available). The funders had no role in study design, data collection and analysis, decision to publish, or preparation of the manuscript.]

Reviewers' comments:

Reviewer's Responses to Questions

**Comments to the Author**

1. Is the manuscript technically sound, and do the data support the conclusions?

Reviewer #1: Partly

Reviewer #2: Yes

2. Has the statistical analysis been performed appropriately and rigorously? 

Reviewer #1: N/A

Reviewer #2: N/A

3. Have the authors made all data underlying the findings in their manuscript fully available?

Reviewer #1: Yes

Reviewer #2: Yes

4. Is the manuscript presented in an intelligible fashion and written in standard English?

Reviewer #1: Yes

Reviewer #2: Yes

5. Review Comments to the Author

Reviewer #1: In this manuscript, Venturi and colleagues describe a useful primer extension technique based on fluorescently labeled primers, with reaction products analyzed by the capillary gel electrophoresis. This method does not require radioactively labeled probes, conventional denaturing polyacrylamide gels, and autoradiography. The described experimental protocol is applicable for the analysis of alternatively processed 5.8S forms and also provides a starting point for a similar type of analysis of other rRNAs. However, from what I see, there is one potentially significant methodological issue concerning the interpretation of key results, and a missing link with the previous literature directly relevant to the RNAs in question. These issues need to be corrected. There are also some statements regarding the processing pathway that are not fully accurate if one considers recent developments in the literature.

Major issues.

1. Lines 49-51: "Two isoforms of 5.8S rRNA have been described: a short form (5.8S short) and a longer one with 5 extra nucleotides at the 5’ of the sequence (5.8S long) (2, 3)".

The authors cite 2010-2011 studies here. I wonder if the authors are familiar with a more recent study, which indicated the presence of 3 isoforms of 5.8S in both mouse and human cells, with two forms seemingly dependent on exonuclease activity (Wang et al., RNA 2015, 21(7):1240-8. doi: 10.1261/rna.051169.115). From looking at the data in Fig 1, it appears the authors are also seeing three isoforms: the electropherogram shows nicely an additional peak, at the expected position for where the 5' end of the 5.8S-cropped is supposed to be, as described by Wang et al (2015). This should be addressed in the discussion of the data, and the Abstract modified as well.

2. Reverse transcriptases are known to efficiently append nontemplated nucleotides to the 3' end of RNA/cDNA duplexes, typically adding a single residue when working on a 5' uncapped RNA template, which is the case here (Biotechniques 2001, 30(3):574-80, 582; Scientific Reports 2017, 7:41769; J Biol Chem. 2019, 294(48):18220–18231). The protocol the authors used does not appear to include blunting of the ends of the RT reaction products. This will affect both the size of the RT products detected in the capillary electrophoresis setting (Fig 1), and the heterogeneity of any detected 5' ends in the sequencing analysis (Fig 2). Because this directly bears on the interpretation of the key results, the authors need, at the very least, discuss this caveat and re-evaluate their data with regard to the precise 5' end identification of different isoforms, taking into account this RT activity.

Minor issues.

1. The principal protocol (line 119 and the following) would be more valuable to other researchers if some additional details were provided to facilitate its application in other labs. In addition to the range of RNA concentrations tested in this study, please indicate the RNA amount you consider optimal for the technique. What kind of tubes did you use to minimize evaporation during incubation of these small volumes? Please indicate the source of the fluorescently labeled primers and the type of purification (if any was used) after the synthesis of these oligos.

2. 48-49: "Ribosome biogenesis has been widely studied in yeast and only a few studies have been conducted on human cell lines." While it is true that this pathway used to be more intensely studied in yeast, there has been steady progress in human cells in recent years, certainly not 'only a few' studies. Consider, for example, the recent work on the human small subunit assembly, no less detailed than any work done in yeast (Sameer Singh et al, Science (2021). DOI:10.1126/science.abj5338).

3. Lines 51-55 "the cleavage at the internal transcribed spacer 1 (ITS1) operated by RNAse MRP", "the long isoform derives from direct ITS1 cleavage by a still uncharacterized enzyme (8)". The model of ITS1 cleavages to which the authors are referring is debatable, and as the recent literature shows, is very likely incorrect, please see Li et al., Int. J. Mol. Sci. 2021, 22(13), 6690; https://doi.org/10.3390/ijms22136690. Nick Watkins' group also showed that MRP deletion had no appreciable effects on ITS1 cleavage in human cells (Ref 7).

5. Refs 13 and 14 are identical.

Reviewer #2: In the manuscript, Venturi and colleagues developed a method based on primer extension and fragment analysis for the peak in the electropherogram to evaluate 5.8S rRNA isoforms. The authors optimized the method by adjusting the RNA and primer amount and confirmed the 5.8rRNA isoforms by reverse transcription and 5’RACE. The results showed that the newly established method was able to analyze the abundance of two isoforms of 5.8S rRNA in samples with small amount or low quality rapidly and safely. It can be a very useful 5.8S rRNA isoform analysis method for basic research and clinical diagnostic applications. The authors provided a clear description for the methodology. However, the manuscript can be improved with some small changes.

Minor concerns

1. In the manuscript, the RNA samples were from cell lines or XpressRef Universal Total RNA. It will be great if the author provide normal patient frozen tissue data to confirm that the method can be used in clinical application.

2. The authors mention that common technique such as Northern Blot requires high quantities of good quality RNA. The authors are recommended to provide the range of RNA amount used in traditional method to show that the new method is more sensitive.

3. Where is Box1 in the Introduction part? I can only find it from the reference paper.

4. In Materials and Methods, line98, the concentration of L-Glutamine and antibiotics should use specified (e.g. 2mM L-Gln) rather than stating 1%.

5. In Materials and Methods, line108, “1X106 cells” 6 should be superscript.

6. A and B labels were missing from Figure1.

7. In Fig 1. legend, it is unnecessary to repeat the details which are already described in method part.

8. There is no statistical analysis in the Materials and Methods. It will be better to do a statistical analysis for figure 4A and B.

6. PLOS authors have the option to publish the peer review history of their article (what does this mean?). If published, this will include your full peer review and any attached files.

Reviewer #1: No

Reviewer #2: No

---

## [Author Response · Author response to Decision Letter 0]

23 Nov 2021

Academic Editor’s comments:

1) Line 104, please specify A260 for RNA quantification.

we added to the text absorbance 260 nm 

2) Please consider adding a diagram in Fig.1 (possibly as Fig.1B and put current Fig.1B as 1C) to illustrate the mechanism of the method, including the labeling of the 5.8S rRNA (three forms) and the 5S rRNA by respective labeled primers in reverse transcription, the separation of the cDNA, the detection and the quantification.

thank you for your suggestion. we preferred to implement Fig.1A with additional details illustrating the mechanism of the method. Fig1B shows a typical electropherogram resulting from its application.

3) Please check sentence 241 for readability.

we changed the sentence: We observed that in two independent experiments 5.8S long/short value is similar in each sample, and that the average ratio value of the ratio results is stable when RNA amount decreases.

4)Fig4B, line 254, please check whether the ratio decreases or rises.

 we changed the approach to calculate the amount of the different 5.8S rRNA isoforms, figures and text has been changed accordingly.

Reviewer #1

major

1. Lines 49-51: "Two isoforms of 5.8S rRNA have been described: a short form (5.8S short) and a longer one with 5 extra nucleotides at the 5’ of the sequence (5.8S long) (2, 3)".

The authors cite 2010-2011 studies here. I wonder if the authors are familiar with a more recent study, which indicated the presence of 3 isoforms of 5.8S in both mouse and human cells, with two forms seemingly dependent on exonuclease activity (Wang et al., RNA 2015, 21(7):1240-8. doi: 10.1261/rna.051169.115). From looking at the data in Fig 1, it appears the authors are also seeing three isoforms: the electropherogram shows nicely an additional peak, at the expected position for where the 5' end of the 5.8S-cropped is supposed to be, as described by Wang et al (2015). This should be addressed in the discussion of the data, and the Abstract modified as well. 

Thank you for the comment. We amended the text, the figure and the bibliography mentioning the additional cropped isoform and included its evaluation in our study. 

2. Reverse transcriptases are known to efficiently append nontemplated nucleotides to the 3' end of RNA/cDNA duplexes, typically adding a single residue when working on a 5' uncapped RNA template, which is the case here (Biotechniques 2001, 30(3):574-80, 582; Scientific Reports 2017, 7:41769; J Biol Chem. 2019, 294(48):18220–18231). The protocol the authors used does not appear to include blunting of the ends of the RT reaction products. This will affect both the size of the RT products detected in the capillary electrophoresis setting (Fig 1), and the heterogeneity of any detected 5' ends in the sequencing analysis (Fig 2). Because this directly bears on the interpretation of the key results, the authors need, at the very least, discuss this caveat and re-evaluate their data with regard to the precise 5' end identification of different isoforms, taking into account this RT activity.

The issue regarding RT-mediated extra nucleotide addition was considered in the discussion (lines 302-308).

Minor

1. The principal protocol (line 119 and the following) would be more valuable to other researchers if some additional details were provided to facilitate its application in other labs. In addition to the range of RNA concentrations tested in this study, please indicate the RNA amount you consider optimal for the technique. What kind of tubes did you use to minimize evaporation during incubation of these small volumes? Please indicate the source of the fluorescently labeled primers and the type of purification (if any was used) after the synthesis of these oligos.

we added these details in the protocol. The optimal RNA amount is 250 ng. tubes multiply-Pro cup 0.2ml PP sarstedt. The source of fluorescently labeled primers is IDT and they have been purified by HPLC. 

2. 48-49: "Ribosome biogenesis has been widely studied in yeast and only a few studies have been conducted on human cell lines." While it is true that this pathway used to be more intensely studied in yeast, there has been steady progress in human cells in recent years, certainly not 'only a few' studies. Consider, for example, the recent work on the human small subunit assembly, no less detailed than any work done in yeast (Sameer Singh et al, Science (2021). DOI:10.1126/science.abj5338).

thank you for the suggestion, we changed the sentence accordingly. 

3. Lines 51-55 "the cleavage at the internal transcribed spacer 1 (ITS1) operated by RNAse MRP", "the long isoform derives from direct ITS1 cleavage by a still uncharacterized enzyme (8)". The model of ITS1 cleavages to which the authors are referring is debatable, and as the recent literature shows, is very likely incorrect, please see Li et al., Int. J. Mol. Sci. 2021, 22(13), 6690; https://doi.org/10.3390/ijms22136690. Nick Watkins' group also showed that MRP deletion had no appreciable effects on ITS1 cleavage in human cells (Ref 7).

We thank the reviewer, we changed the text in the introduction and in the box1 to show how this specific issue is currently under debate. 

4. Refs 13 and 14 are identical.

We corrected the error. 

Minor concerns

1. In the manuscript, the RNA samples were from cell lines or XpressRef Universal Total RNA. It will be great if the author provide normal patient frozen tissue data to confirm that the method can be used in clinical application.

We agree with the reviewer on the importance of the issue. We considered that using a limited number of pathological samples could not provide a definitive confirmation of the possibility to use the method on clinical samples. To test the method in controlled conditions we preferred to perform the experiment reported in Figure 4C.

2. The authors mention that common technique such as Northern Blot requires high quantities of good quality RNA. The authors are recommended to provide the range of RNA amount used in traditional method to show that the new method is more sensitive.

We reported RNA concentration typically used in northern blot analysis in the discussion paragraph. 

3. Where is Box1 in the Introduction part? I can only find it from the reference paper.

The text of box1 is reported in the text file while the image is uploaded as a figure.

4. In Materials and Methods, line98, the concentration of L-Glutamine and antibiotics should use specified (e.g. 2mM L-Gln) rather than stating 1%.

We modified the text specifying the exact concentration of L-Glutamine, Penicillin and Streptomycin. 

5. In Materials and Methods, line108, “1X106 cells” 6 should be superscript.

We corrected the error. 

6. A and B labels were missing from Figure1.

We corrected the figure.

7. In Fig 1. legend, it is unnecessary to repeat the details which are already described in method part.

We modified the figure legend accordingly.

8. There is no statistical analysis in the Materials and Methods. It will be better to do a statistical analysis for figure 4A and B.

We added paragraph to the Materials and methods indicating the statistical analysis performed. We included statistical analysis in figure 4A and B.

---

## [Editor Report · Decision Letter 1]

29 Nov 2021

PONE-D-21-30501R1Primer extension coupled with fragment analysis for rapid and quantitative evaluation of 5.8S rRNA isoformsPLOS ONE

Dear Dr. Montanaro,

Thank you for submitting your manuscript to PLOS ONE. After careful consideration, we feel that it has merit but does not fully meet PLOS ONE’s publication criteria as it currently stands. Therefore, we invite you to submit a revised version of the manuscript that addresses the points raised during the review process.

Legend for Fig.1A should define the short names for the three isoforms.

Line 28 and 76, ”two 5.8S isoforms” should be “three 5.8s isoforms”.

We look forward to receiving your revised manuscript.

Kind regards,

Baisong Lu, Ph.D

Academic Editor

PLOS ONE
---

## [Author Response · Author response to Decision Letter 1]

30 Nov 2021

Academic Editor’s comments:

Legend for Fig.1A should define the short names for the three isoforms.

We amended the figure legend

Line 28 and 76, ”two 5.8S isoforms” should be “three 5.8s isoforms”.

We corrected the errors

---

## [Editor Report · Decision Letter 2]

3 Dec 2021

Primer extension coupled with fragment analysis for rapid and quantitative evaluation of 5.8S rRNA isoforms

PONE-D-21-30501R2

Dear Dr. Montanaro,

We’re pleased to inform you that your manuscript has been judged scientifically suitable for publication and will be formally accepted for publication once it meets all outstanding technical requirements.

Kind regards,

Baisong Lu, Ph.D

Academic Editor

PLOS ONE
---

## [Editor Report · Acceptance letter]

7 Dec 2021

PONE-D-21-30501R2 

Primer extension coupled with fragment analysis for rapid and quantitative evaluation of 5.8S rRNA isoforms 

Dear Dr. Montanaro:

I'm pleased to inform you that your manuscript has been deemed suitable for publication in PLOS ONE. Congratulations! Your manuscript is now with our production department. 

Kind regards, 

on behalf of

Dr. Baisong Lu 

Academic Editor

PLOS ONE